# Association between IGF-1 and IGFBPs in Blood and Follicular Fluid in Dairy Cows Under Field Conditions

**DOI:** 10.3390/ani14162370

**Published:** 2024-08-15

**Authors:** Christina Schiffers, Idil Serbetci, Kirsten Mense, Ana Kassens, Hanna Grothmann, Matthias Sommer, Christine Hoeflich, Andreas Hoeflich, Heinrich Bollwein, Marion Schmicke

**Affiliations:** 1Veterinary-Endocrinology and Laboratory Diagnostics, Clinic for Cattle, University of Veterinary Medicine Hannover, Foundation, 30173 Hannover, Germany; marion.schmicke@tiho-hannover.de; 2Clinic of Reproductive Medicine, Vetsuisse Faculty, University of Zurich, 8057 Zurich, Switzerland; idil.serbetci@uzh.ch (I.S.); heinrich.bollwein@uzh.ch (H.B.); 3SYNETICS Germany GmbH, 27283 Verden, Germany; kirsten.mense@synetics.world (K.M.); ana.kassens@synetics.world (A.K.); hanna.grothmann@synetics.world (H.G.); 4Agrargenossenschaft Helmsdorf eG, 06347 Gerbstedt, Germany; info@ag-helmsdorf.de; 5Ligandis UG, 18276 Gülzow-Prüzen, Germany; christine.hoeflich@ligandis.de; 6Research Institute for Farm Animal Biology (FBN), 18196 Dummerstorf, Germany; hoeflich@fbn-dummerstorf.de

**Keywords:** IGF-1, IGFBP, somatotropic axis, dairy cow, cattle, fertility, metabolic status

## Abstract

**Simple Summary:**

In dairy cows, a hormone called insulin-like growth factor 1 (IGF-1) helps to control growth and reproduction. It affects how oocytes develop in the ovaries. IGF-1 is usually attached to binding proteins (IGFBPs), which can limit its effects. This study looked at how IGF-1 transfers between blood and the fluid around the oocyte. We checked IGF-1 and protein levels in both places and also measured energy levels using ketones. We examined gene activity in ovary cells too to look for local regulation. We found that IGF-1 levels in blood and ovarian fluid are related. High ketone levels, which indicate low energy, were linked to lower IGF-1 and some of the IGFBP. Some gene activity is also related to IGF-1 levels. In summary, while additional research is necessary to fully understand the local regulatory mechanisms, our findings suggest that the transfer of IGF-1 between systems may link metabolic issues to fertility challenges in dairy cows.

**Abstract:**

Insulin-like growth factor 1 (IGF-1) regulates dairy cow reproduction, while the paracrine IGF system locally influences fertility. In both systems, IGF-1 bioactivity is regulated through binding proteins (IGFBPs) inhibiting IGF-1 binding to its receptor (IGF1R). This study aimed to investigate a possible transfer between this endocrine and paracrine system. Therefore, blood and follicular fluid (FF) from postpartum dairy cows were analysed for ß-hydroxybutyrate (BHB), IGF-1, IGFBP-2, -3, -4, -5, and an IGFBP fragment in two study parts. The mRNA expression of *IGFBP-2*, *IGFBP-4*, *IGF1R*, and the pregnancy-associated plasma protein A (*PAPP-A*) in granulosa cells was measured. The results showed correlations between plasma and FF for IGF-1 (r = 0.57, *p* < 0.001) and IGFBP-2 (r = −0.57, *p* < 0.05). Blood BHB negatively correlated with IGF-1 in blood and FF and IGFBP-3, -5 and total IGFBP in blood (IGF-1 plasma: r = −0.26, *p* < 0.05; FF: r = −0.35, *p* < 0.05; IGFBP-3: r = −0.64, *p* = 0.006; IGFBP-5: r = −0.49, *p* < 0.05; total IGFBP: r = −0.52, *p* < 0.05). A negative correlation was found between *IGFBP-2* expression and IGF-1 concentration in FF (r = −0.97, *p* = 0.001), while an IGFBP fragment positively correlated with *IGF1R*-mRNA in FF (r = 0.82, *p* = 0.042). These findings suggest a transfer and local regulation between the somatotropic axis and the follicular IGF system, linking the metabolic status with local effects on dairy cow fertility.

## 1. Introduction

The somatotropic axis influences physiological processes in the entire organism, especially affecting the reproduction of the lactating dairy cow [1,2,3]. In this system, the hypothalamic growth hormone (GH) affects tissues by its receptor (GHR). In many organs, including reproductive tissue and the liver, GH induces the synthesis of insulin-like growth factor 1 (IGF-1) and some of its specific binding proteins (IGFBP) [4]. IGF-1 exerts its effects by binding to two specific receptors (IGF1R and IGF2R), leading to intracellular signalling that regulates cell growth, proliferation, and differentiation [5,6]. The IGF1R is present in various tissues, including granulosa and theca cells (GC, TC) in the ovarian follicle [7,8]. The IGFBPs (IGFBP-1 to -6) control IGF-1 by preventing its binding to the IGF1R [9]. This is additionally altered by enzymes like the pregnancy-associated plasma protein A (PAPP-A) that proteolyse IGFBP to release IGF-1 [10]. These regulating mechanisms are key components of both the somatotropic axis and the paracrine IGF system in the follicle of the cow.

The local effect of IGF-1 on the follicle has been demonstrated in several studies. These studies showed that IGF-1 promotes follicular growth and selection as well as higher developmental competence and quality of the oocyte [11,12,13,14]. Zhou et al. [15], observed that IGF-1 increases the sensitivity of the follicle to follicle-stimulating hormone (FSH) and luteinising hormone (LH), thereby influencing follicle selection. Further in vitro experiments showed that the addition of IGF-1 increased the expression of LH receptor mRNA in the bovine GC culture and estradiol synthesis in GC from small (6–8 mm) and large follicles (≥9 mm; [16]). These findings highlight the importance of the local IGF-1 concentration in follicular fluid (FF) for reproductive processes. 

The local concentration of IGF-1 may have its origin in either GC production or serum filtration. The precise mechanism of local IGF-1 synthesis within the follicle in cattle remains unclear. In vitro studies have observed the synthesis of IGF-1 mRNA in GC from bovine follicles of varying sizes [17,18]. Other studies observed minimal expression of *IGF-1* mRNA and IGF-1 levels in cell media during in vitro experiments with cow GC or found no expression at all [19,20,21]. Additionally, follicular fluid is also regarded as a filtrate of the serum [22,23,24]. Thus, it may be postulated that a transfer from the somatotropic axis to the IGF system in the follicle represents another potential source of follicular IGF-1. Considering this, fluctuations in serum IGF-1 and IGFBP levels during the postpartum period reflect somatotropic adjustments to lactation demands. These fluctuations are often the result of the animals experiencing a negative energy balance (NEB) and liver metabolism overload, as indicated by high β-hydroxybutyrate (BHB) levels in the blood [25,26,27]. 

In light of the aforementioned findings, a hypothesis emerges that fluctuations in serum IGF-1 and IGFBP levels, reflecting somatotropic adjustments to early lactation metabolic demands, may influence follicular IGF-1 dynamics and subsequently impact dairy cow fertility. To elucidate this relationship and assess the implications for reproductive outcomes, our study aimed to investigate the transfer between the somatotropic axis and the follicular IGF system, thereby illuminating its role in linking metabolic status and fertility in dairy cows.

## 2. Materials and Methods

The objective of this study was to examine the concentration of IGF-1 and IGFBP in both blood and FF, with the aim of establishing a potential association between the somatotropic axis and the follicular IGF system. Consequently, blood samples were collected at two or three time points postpartum (p.p.), and FF was aspirated at one time point. Furthermore, in order to assess the potential influence of metabolic processes, the concentration of BHB in blood was measured in each sample. Additionally, the expression of specific genes in the obtained GC was analysed in order to take potential local regulatory mechanisms into account. The samples and data used in this study were collected in two different study parts (study parts I and II, Table 1).

### 2.1. Animals

**Study part I** included Holstein dairy cows (1st to 9th lactation) kept by the Agricultural cooperative Helmsdorf in Gerbstedt, Saxony-Anhalt, Germany. All lactating animals were maintained in a freestall barn system with an outdoor climate. First lactation animals were maintained in a group within the barn section, which was equipped with high stalls on slatted floors with automated slider manure removal (LJM Agro, Højmark, Denmark) and straw bedding. Cows that had undergone two lactations or more were maintained in a separate group within a distinct section of the barn. This group was housed in high stalls, provided with straw bedding, and equipped with a slated floor and slider manure removal system. The farm used a 40-head outdoor milking rotary (built in 2014, Lemmer-Fullwood GmbH, Lohmar, Germany) for milking three times daily. The animals were provided with a total mixed ration comprising maize and lucerne silage, grain and cob mix, rapeseed meal, pressed pulp, draff, finely ground barley, molasses, and a mineral feed mixture produced and formulated by the company AHRHOFF, (Bönen, Germany). The two groups received identical rations. Sample collection in both study parts I and II was adapted to the farm routine, and all animals were accustomed to being handled by humans as well as the husbandry and feeding conditions on each farm.

For **study part II**, 25 blood samples, the corresponding pooled follicular fluid from the same animal, and the corresponding data (age, lactation, BCS) were provided by the Department of Assisted Reproduction at the Clinic for Reproductive Medicine at the Faculty of Veterinary Medicine of the University of Zurich, Switzerland. Study part II included Holstein dairy cows kept on AgroVet Strickhof Group farms in Switzerland. Those animals were examined and sampled at regular intervals before and after calving as part of another project [28]. The study animals were selected from a group of cows that were subject to rotary or automated milking. The two groups were maintained in cubicle pens with deep stalls and slatted or level floors. The manure was removed using an automated manure scraper, and the animals were provided with access to an exercise yard. The stables were constructed between 2014 and 2018 and are equipped with climate control facilities.

The robot group was provided with a total mixed ration comprising maize silage, grass silage, hay, lucerne meal, maize meal, wheat straw, and AgroVet 2020 concentrate. The animals in the robotic milking system were provided with supplementary concentrated feed. The total mixed ration of the animals in the rotary milking group consisted of maize silage, grass silage, hay, AgroVet 2020 concentrate, and the feed supplement UFA 249 Prima (UFA AG, Herzogenbuchsee, Switzerland).

### 2.2. Study Design and Sample Collection

**Study part I:** Animals underwent a general and gynaecological examination at all time points (30 ± 2 days p.p.; 50 ± 3 days p.p.; 65 ± 16 days p.p.). The condition of the uterus and ovaries was assessed by transrectal palpation and ultrasound examination (linear probe, 4.5–8.5 MHz, BL Tecnoscan, Scan4Animal, Wuppertal, Germany), and the cycle status was determined. Blood samples were taken, and the Body Condition Score (BCS) of the animals was documented. Based on the results of these examinations, 25 animals were selected for subsequent oestrus synchronisation. Oestrus synchronisation was performed by inducing luteolysis twice at 14-day intervals (intramuscular injection of 0.15 mg cloprostenol; 2 mL Ovaren^®^, Ceva Tiergesundheit GmbH, Düsseldorf, Germany). Following oestrus synchronisation, transrectal ultrasound examinations were performed on the animals to determine functional structures on their ovaries. Animals that showed regression of the corpus luteum and simultaneous follicle growth after the second injection were selected for transvaginal ultrasound-guided follicle puncture (ovum pickup, OPU) on the third day after the second injection of prostaglandins. OPU was then performed on 17 animals, and simultaneously blood samples were taken. During OPU, a total of 27 follicles (diameter 5–20 mm) were punctured and FF was aspirated. Granulosa cell (GC) collection was possible for a total of eight follicles. After centrifugation and separation of FF and GC, lysis buffer (Table A1) was added to the GC, and both GC and FF were stored in liquid nitrogen until analysis.

**Study part II:** For the project of Serbetci et al. [28], pregnant animals (2nd to 8th lactation) were observed and sampled between the point of drying off (approx. 8 weeks antepartum; a.p.) and 8 weeks p.p. In weeks 5, 6, 7, and 8 p.p., oocytes and cumulus cells were collected twice a week using OPU for subsequent in vitro experiments and analyses. For the study described here, FF collected during week 5 p.p. as well as serum and plasma samples from 25 animals collected in the same week and in week 8 p.p. were made available for study part II, stored at −80 °C, and used for the laboratory analyses (Table 1). In study part II, follicles with a diameter greater than 3 mm were aspirated; the individual diameter of each follicle was not recorded, and FF was pooled for all follicles within an animal.

### 2.3. Laboratory Analysis

#### 2.3.1. Blood Beta Hydroxybutyrate (BHB)

The concentration of BHB was determined in serum samples using a handheld BHB-measuring device (in mmol/L, range 0.1–8.0 mmol/L, BHB-Check, TaiDoc Technologie Corporation, New Taipei City, Taiwan). The device has been used and validated in bovine blood in previous studies [29,30,31,32]. In bovine plasma, Bach et al. [32] found a coefficient of variation (CV) between 2.0 and 7.9% for the device, and Leal Yepes et al. [29] reported a CV of 11% for bovine plasma at room temperature. To verify the device’s suitability for use in bovine serum, the results were compared with those obtained from photometric laboratory analysis (Pentra C400, Horiba Medical, Montpellier, France; r = 0.994; *p* < 0.0001; [31]). 

#### 2.3.2. IGF-1

The total concentration of IGF-1 (free and bound) in plasma and follicular fluid was quantified using a commercial sandwich enzyme-linked immunosorbent assay (ELISA) for the quantitative detection of human IGF-1 (Mediagnost GmbH, Reutlingen, Germany). In this assay, IGF-1 is initially dissociated from the IGFBP. To this end, the plasma is diluted 1:21 and the follicular fluid 1:10 with an acidic sample buffer (pH < 3.1). Consequently, only total IGF-1 was quantified in this study, and the term IGF-1 is henceforth employed to denote the sum of bound and unbound IGF-1. The ELISA was previously validated for the quantitative detection of bovine IGF-1 [31]. In order to validate the ELISA, Grone et al. (2022, [31]) compared the concentrations of IGF-1 in bovine plasma measured with the ELISA with those measured with a radioimmunoassay that had been used in several older studies [31,33,34,35] (r = 0.97; *p* < 0.0001; r^2^ = 0.95, *p* < 0.0001; ELISA: intra-assay coefficient of variation 5.1%, inter-assay coefficient of variation 9.3%). 

#### 2.3.3. IGFBP

The presence of IGFBP in plasma and FF was determined through the use of quantitative Western ligand blotting performed by Ligandis UG (Gülzow-Prüzen, Germany). The same methodology previously described in the literature was employed for the analysis of plasma and FF [36,37,38]. In summary, this procedure employs biotin-labelled human insulin-like growth factor 2 (IGF-2) to detect the IGFBP, following their separation through gel electrophoresis. To mitigate the impact of positional effects on the gel, the quantification of IGFBP was conducted in relation to an IGFBP-3 standard, which was applied alternately to the samples (Figure 1). The standard comprised a fixed quantity of 1.5 ng recombinant human IGFBP-3 (Accession No. CAA46087, R&D Systems, Inc., Minneapolis, MN, USA). Thereafter, the quantity of IGFBP in the samples was calculated in relation to the standard (=100%) on the two adjacent rows. The allocation and utilisation of the IGFBP-3 standard are shown in Figure 1.

#### 2.3.4. Gene Expression

GC were obtained from eight different follicles derived from six different animals in study part I. Cells were aspirated and prepared from five follicles with a diameter of ≥15 mm (maximum diameter 20 mm) and from three follicles with a diameter of <15 mm (minimum diameter 9 mm; Table A2). The RNA was isolated using an established TRIzol^TM^ protocol, as outlined by Stiensmeier (2021) [39]. For quantification, a NanoDropTM spectrophotometer was used (NanoDrop 2000c, ThermoFisher Scientific, Inc., Waltham, MA, USA). Subsequently, the RNA was stored at −20 °C for further analysis. Only samples in which sufficient RNA could be isolated (≥15 ng/μL, as detailed in Appendix A, Table A2) were used for the synthesis of complementary deoxyribonucleic acid (cDNA) and RT-qPCR. The samples were then adjusted to a concentration of 20 ng/μL, and 10 μL of reaction mix A (Table A3) was added to the RNA. Thereafter, the RNA was denatured in a thermal cycler. Next, 5 μL of reaction mix B (Table A4) were added, and the mixture was heated to 37 °C for 15 min to facilitate primer attachment. This was followed by a further 45 min at 42 °C for cDNA synthesis and 10 min at 70 °C for enzyme inactivation. The synthesised cDNA was stored at −20 °C until further use.

Quantitative real-time reverse transcriptase–polymerase chain reaction (RT-qPCR) was applied to analyse the gene expression of the GC. The primers were selected based on previous work, and the sequences were adopted (Table 2). The correctness of the sequences was verified through BLAST-Search. The mean value of GAPDH (glyceraldehyde-3-phosphate dehydrogenase) and 18S rRNA (18S ribosomal RNA) was used as the reference (RG) [40,41,42]. All primers had been previously tested in RT-qPCR runs to ascertain their suitability for use with cDNA derived from RNA isolated from GC. Primers were deemed suitable if their cycle threshold (Ct) value fell within the range of 10 to 30, and if there were a minimum of 10 cycles between the Ct value of the sample utilised and that of the negative control in the test run.

In the event that the melting curve for a given primer pair exhibited a single peak and no additional peaks attributable to primer dimers or impurities, the relative amount of RNA and the ratio of mRNA expression were calculated using the Ct value of the sample (mean value from double determination) and the ΔΔCt method in relation to the combination of both reference genes. For the calculation of ΔΔCt, the mean value of ΔCt for the respective primer pair was formed from all granulosa cell samples (*n* = 6) and used as a calibrator. 

(${Ct}_{RG}={(Ct}_{18SrRNA}+{Ct}_{GAPDH})/2;∆Ct={Ct}_{Gene}-{Ct}_{RG};∆∆Ct=∆Ct-∆{Ct}_{calibrator};\;Ratio=2^{-∆∆Ct};$ [40,43,44]). 

**Table 2 animals-14-02370-t002:** Information on the primers used for RT-qPCR.

Gene	Accession Number	Primer Sequence		Literature
*GAPDH*	NM_ 001034034.1	Forward	CAA CAT CAA GTG GGG TGA TG	[45]
		Reverse	GGC ATT GCT GAC AAT CTT GA	
*18S rRNA*	NR_ 036642.1	Forward	ACC CAT TCG AAC GTC TGC CCT ATT	[39]
		Reverse	TCC TGG GAT GTG GTA GCC GTT TCT	
*IGFBP-2*	AF074854	Forward	GAC GGG AAC GTG AAC TTG ATG	[41]
		Reverse	TCC TTC ATG CCG GAC TTG A	
*IGFBP-4*	NM_ 174557.3	Forward	CCC AAG TCT GTG GGA GAA GA	[45]
		Reverse	AAG GAC CTG GGG AGG AGT AA	
*PAPP-A*	XM_ 613511.6	Forward	TGG AGA ACG CTT CGC TCA ACT G	[46]
		Reverse	ACG CTG GGT CCT GTC TGG CTT T	
*IGF1R*	XM_ 606794.3	Forward	CCA AAA CCG AAG CTG AGA AG	[45]
		Reverse	TCC GGG TCT GTG ATG TTG TA	

*GAPDH* = glyceraldehyde 3-phosphate dehydrogenase; *18S rRNA* = 18S ribosomal RNA; IGFBP-2 = insulin-like growth factor binding protein-2; *IGFBP-4* = insulin-like growth factor binding protein-4; *PAPP-A* = pregnancy-associated plasma protein A; *IGF1R* = insulin-like growth factor 1 receptor.

### 2.4. Statistical Analysis

For statistical evaluation, the blood and FF samples from both study parts were combined according to the time of collection. Prior to analysing the samples of study parts I and II together for parameters (IGF-1 and BHB), it was first ascertained whether there were significant differences between the samples (Table A5). If the results from study parts I and II were analysed separately, this is stated in the figure descriptions.

The distribution of all laboratory results and ovarian findings was evaluated using the Shapiro–Wilk and Kolmogorov–Smirnov tests. Data that met the criteria for normal distribution are presented as mean ± standard deviation (SD), while data that did not meet these criteria are presented as median/95%/5% percentile. In instances where the data were normally distributed, a paired or unpaired Student’s t-test was conducted. In the event of more than two sample groups, a one-way analysis of variance (ANOVA) with a Tukey test as a post hoc test was conducted. In the case of non-normally distributed values at individual time points or in blood or follicular fluid, two paired samples were analysed with the Wilcoxon signed-rank test, unpaired samples with the Mann–Whitney test, and more than two samples with a Kruskal–Wallis test for unpaired samples. Subsequently, a Dunn’s test was employed as a post hoc test following the Kruskal–Wallis test. The relationships between the various parameters were investigated using Spearman’s correlation coefficient for non-normally distributed data and Pearson’s correlation coefficient for normally distributed data. In instances where a significant correlation was identified, simple linear regression analysis was conducted as an additional measure. In cases where regression analysis has been conducted, the dependent variable is plotted on the y-axis and the independent variable on the x-axis. If significant linear regression has been identified, regression lines are presented in the figures. To ensure the accuracy and reliability of the correlation and regression analysis, any outliers were excluded. Outliers were identified using the ROUT method (robust regression and outlier removal) in GraphPad Prism (version 9.4.0) with a Q value of 1% [43]. If data were excluded from the correlation and/or regression analysis due to this reason, this is indicated in the text or figure caption. A *p*-value of <0.05 was considered statistically significant.

## 3. Results

### 3.1. IGF-1

**Study part I:** In study part I, IGF-1 in blood and FF did not differ significantly (123.9/246.8/44.0 ng/mL vs. 112.9 ± 26.6 ng/mL, *p* > 0.05). Furthermore, there were no significant differences for IGF-1 in FF depending on the diameter of the follicle (diameter 5–20 mm). 

**Study part II:** IGF-1 concentrations were higher in blood compared to FF in study part II (134.2 ± 44.8 ng/mL, *n* = 25 vs. 67.7 ± 32.2, *n* = 24; *p* < 0.0001). 

**Comparison of study parts I and II:** IGF-1 in follicular fluid was higher in study part I than in study part II (112.9 ± 26.6 ng/mL vs. 67.7 ± 32.2 ng/mL, *p* < 0.0001). 

**Study parts I and II combined:** IGF-1 in blood and FF correlated significantly with each other for both study parts combined (Figure 2A). BHB in blood correlated negatively with IGF-1 in blood and in FF in both study parts combined (Figure 3D,E).

### 3.2. IGFBP

**Study part I:** On day 65 ± 16 p.p. in blood and FF, IGFBP-1, -2, -3, -4, -5, and a suspected IGFBP fragment (IGFBP-F) could be found (Table 3). The pattern of IGFBP is shown in Figure 4. In blood, the proportion of IGFBP-2 in total IGFBP concentration was higher than in FF (30.9/53/21% vs. 13.8/36.3/0.5%, *p* < 0.001). For IGFBP-F and IGFBP-5, the percentage was higher in FF than in plasma (IGFBP-F: 10.27 ± 3.18% vs. 5.22 ± 2.31%, *p* < 0.001; IGFBP-5: 21.82 ± 8.43% vs. 13.77 ± 3.04%, *p* = 0.001). Regarding the association between blood and FF, the concentrations of IGFBP-2 correlated with each other (Figure 2B). For IGFBP in FF, there was no difference depending on the diameter of the follicle. In FF with increasing follicle diameter, the proportion of IGFBP-3 in total IGFBP increased and that of IGFBP-4 decreased (IGFBP-3: *n* = 27, follicle diameter 5–20 mm, r = 0.48; r^2^ = 0.23; IGFBP-4: *n* = 26, one outlier eliminated with IGFBP-4 share: 33.35%, follicle diameter 5–20 mm, r = −0.47; r^2^ = 0.22; *p* < 0.05). Further IGFBP-3, IGFBP-5, and total plasma IGFBP concentration correlated negatively with BHB (Figure 3A–C).

### 3.3. Gene Expression 

**Study part I:** The relative gene expression levels in the GC extracted from the aforementioned follicles are presented in Table 4. A negative correlation was observed between the ratio of *IGFBP-2* mRNA expression and the concentration of IGF-1 in the corresponding follicular fluid (FF). Further, a positive correlation was found between *IGF1R* mRNA expression and the concentration of IGFBP-F in the FF (Figure 5). No significant associations were identified between *PAPP-A* mRNA expression and either follicle size, IGF-1, or IGFBP concentration.

## 4. Discussion

### 4.1. IGF-1 in Plasma and Follicular Fluid

The objective of this study was to examine the potential associations between the somatotropic axis and the IGF system in follicles of dairy cows, with a view to identifying a possible link between the animal’s metabolic status and its reproductive performance. A correlation between plasma and FF was identified for IGF-1, as previously documented in the literature [47,48,49]. This finding is consistent with the studies of Perks et al. [19] and Armstrong et al. [21] which demonstrated that bovine follicles produce only very low or no levels of IGF-1. This is in contrast with the results of Schams et al. [17], Yuan et al. [18], and Spicer et al. [50], who detected in vitro mRNA expression of IGF-1 in the follicle [17,18,50]. Moreover, Spicer et al. [51], also presented an argument against the exclusively systemic origin of IGF-1. In their study, short-term fasting of heifers (24 and 48 h) resulted in a reduction in IGF-1 in the blood but not in the FF. Conversely, a study by Chase et al. [52], demonstrated that cattle with GHR deficiency and consequently reduced systemic IGF-1 exhibited diminished follicle development, while FSH, LH, and oestradiol concentrations remained unaltered. Furthermore, both IGF-1 in plasma and FF were reduced following the immunisation of cattle against growth hormone-releasing hormone (GHRH) [53,54], whereas Cohick et al. [54] were unable to detect any change in the mRNA expression of *IGF-1*, *GHR*, or *IGFBP-2*, *IGFBP-3*, and *IGFBP-4* in the follicles of such animals. Therefore, when the results of this and the above studies are combined, it is not possible to fully confirm the hypothesis that IGF-1 in the follicle is exclusively of systemic origin. However, from these results it can still be hypothesised that systemic and GH-dependent changes in the somatotropic axis are to some extent reflected in the follicle, while a form of counter-regulation takes place that is not dependent on systemic GH and for which it is still unclear whether it involves paracrine synthesis of IGF-1.

In the FF, no statistically significant differences were observed in IGF-1 levels in relation to the size of the follicle. This was irrespective of whether the follicles were classified according to their size or in relation to the size of other follicles in the same animal. Similar observations were made by Stewart et al. [8], who did not detect differences in IGF-1 concentrations in dominant, small (<6 mm) or large, non-dominant follicles. Nevertheless, the findings of Stewart et al. [8] indicated a reduction in IGFBP-2 and an increased binding of LH in dominant follicles. Echternkamp et al. [55], observed higher concentrations of IGF-1 in large follicles that synthesised oestrogen compared to those that synthesised less oestrogen. The discrepancy between the aforementioned results and those of the present study may be attributed to the fact that Echternkamp et al. (1994) classified follicles according to their oestradiol synthesis. A comparable classification of follicles was not conducted in the present study, as the majority of punctures were performed on pre-ovulatory follicles. An additional potential explanation for the absence of observed differences in IGF-1 levels between follicle sizes in the present study and that of Stewart et al. (1996) is ref. [8], that it is not the overall quantity of IGF-1 that is decisive for dominant follicle growth, but rather the proportion of free, receptor-accessible IGF-1 and, consequently, the amount of IGFBP.

### 4.2. IGFBP in Plasma and Follicular Fluid

The role of IGFBPs within the IGF system is to regulate bioactivity through the inhibition of IGF-1 receptor binding, thereby exerting a pivotal influence over the follicle and, consequently, reproduction. The present study revealed the presence of IGFBP-2, IGFBP-3, IGFBP-4, and IGFBP-5 in all samples obtained from study part I. Furthermore, a protein band with a molecular weight below that of IGFBP-4 (24 kDa) was detected [56,57]. This IGFBP pattern is consistent with the results of several studies that also identified these binding proteins in plasma or follicular fluid [58,59,60,61]. The comparably high level of IGFBP-3 in plasma and FF indicates that IGFBP-3 functions as a systemic and local IGF storage, thereby prolonging its half-life. With regard to IGFBP-3 as the principal component in plasma, Grimard et al. (2013, [62]) demonstrated comparable outcomes in lactating dairy cows. In follicular fluid (FF), IGFBP-3 was the IGFBP with the highest concentration in this study and in other studies referenced [55,58,61]. The results of Funston et al. (1996, [61]), which demonstrate a positive correlation between IGFBP-3 and IGF-1 in the follicle, lend support to this assumption. Nevertheless, further investigation is required to ascertain the source of IGFBP-3 in the FF and to determine whether and how its concentration in the follicle fluctuates under varying conditions.

In this study, the concentration of IGFBP-2 in the largest follicle and the mean value from all punctured follicles were found to correlate with that in plasma. The observed correlation indicates that IGFBP-2 in the follicle has systemic implications, predominantly originating from the hepatic system. In contrast to IGF-1, the mRNA expression of *IGFBP-2* in the GC of cattle has been clearly demonstrated in several studies [41,63,64,65]. In this study, mRNA expression of *IGFBP-2* was also detected in the obtained GC. This indicates that despite the correlation shown between plasma and FF, IGFBP-2 in the follicle is not exclusively of systemic origin and is also synthesised locally. It may, however, be postulated that the observed correlation is indicative of a role for IGFBP-2 in the general transport of IGF-1 into different tissues [66,67]. The lack of a significant correlation between the IGFBP-2 concentration in the FF and its mRNA, as reported by Santiago et al. (2005, [64]), lends support to the hypothesis that IGFBP-2 originates from the blood circulation in addition to its synthesis in the GC.

While other studies have reported significantly lower levels of IGFBP-2 in dominant follicles compared to other punctuated follicles, this study did not find any significant differences in the levels of IGFBPs in the FF between follicle sizes [55,68,69]. As previously stated, this may be due to the inability to clearly distinguish between dominant and non-dominant follicles. The results of our study revealed a negative correlation between the proportion of IGFBP-4 in total IGFBP and a positive correlation between the share of IGFBP-3 and follicle size. Spicer et al. (2001, [69]) demonstrated that the dominant follicle exhibited greater proteolysis of IGFBP-4, which may explain the negative correlation observed for the proportion of IGFBP-4 in the present study. This is in accordance with the findings of Mazerbourg et al. (2001, [10]), who demonstrated that *PAPP-A* mRNA expression in the follicle exhibited a positive correlation with the expression of aromatase and, consequently, with oestrogen synthesis, as well as with the proteolysis of IGFBP-4. Rivera and Fortune (2001, [70]) showed that the formation of more than one dominant follicle in cattle is triggered by the FSH-dependent cleavage of IGFBP-4, among other factors. The studies conducted by Spicer et al. (2001, [69]), Mazerbourg et al. (2001, [10]), and Rivera and Fortune (2001, [70]), in addition to the aforementioned correlation, provide evidence to support the hypothesis that the increased proteolysis of IGFBP-4 represents a mechanism for the local regulation of the IGF system during follicular growth. The local IGF storage function of IGFBP-3, as previously described, is presumably the reason for the proportion of IGFBP-3 increasing at the same time. It is important to note that our study is limited in its ability to fully describe the influence of the IGF system during follicle differentiation. The majority of the follicles punctured were large and presumably pre-ovulatory and were only divided according to their size. Further studies are required to investigate the IGF system at different time points and in more precisely differentiated and categorised follicles.

### 4.3. Association between Metabolism and IGF-1/IGFBP

The present study revealed a negative correlation between IGF-1 and BHB at day 30 ± 2 p.p., a finding that is supported by the results of previous studies [71,72]. The concentration of IGF-1 in the FF was also found to correlate with that of BHB in the serum on the day of OPU. No studies were identified that had also observed this latter correlation. However, Forde et al. [73], demonstrated that serum BHB and NEFA concentrations were higher and insulin, glucose and IGF-1 concentrations were lower in lactating cows compared to non-lactating cows. At the same time, in the three aforementioned studies the composition of amino acids in the FF of preovulatory follicles differed significantly between these animals. This, in conjunction with the correlation between BHB and IGF-1 in plasma and the FF as described in this study, indicates that the metabolic status of the dairy cow following calving exerts an influence on both the somatotropic axis and local components of the IGF-1 system.

This metabolic influence is further reinforced by the observed negative correlation between BHB in serum and IGFBP-3, IGFBP-5, and total IGFBP in plasma. This observation is consistent with those of other studies [25,45,74]. During the period preceding and immediately following parturition, the metabolism of the animal is significantly disrupted, resulting in elevated blood levels of BHB [75]. The findings of this study lend support to the hypothesis that a reduction in the large binding proteins and the total binding capacity in the blood of the animal leads to a reduction in the systemic IGF reservoir in the event of metabolic stress. A shift towards the transport of IGF-1 into the tissue via IGFBP-2, as found in the aforementioned studies, was not observed in this study. Further investigation is required to determine whether this is a general phenomenon. To this purpose, it would be advisable to examine BHB and IGFBP-2 in the blood of additional animals at more time points postpartum than those considered in this study. 

### 4.4. Gene Expression in the Follicle

The inverse relationship between IGF-1 and *IGFBP 2* messenger RNA (mRNA) expression observed in this study differs from the in vitro observations of Voge et al. [76], who observed an increase in *IGFBP-2* expression in granulosa cells (GCs) of dominant follicles following the addition of IGF-1. One potential explanation for this disparate observation of IGF-1 and IGFBP-2 synthesis is the hypothesis proposed by Walters et al. [65], which suggests that the impact of IGF-1 on *IGFBP-2* mRNA expression is contingent upon the follicular stage and the concentration of IGF-1. It is thus plausible that the discrepancies between the findings of the present study and those of Voge et al. [76] can be attributed to the fact that the gene expression in both studies was assessed in GC from differing follicular stages and under varying levels of IGF-1. In light of the aforementioned correlation between IGFBP-2 in plasma and FF, it is plausible to suggest that following the saturation of the IGF-1 binding capacity of IGFBP-3 in the blood, systemic IGFBP-2 may increase in order to facilitate the transport of IGF-1 into the tissue. In parallel, an elevated total concentration of IGF-1 diminishes the local synthesis of IGFBP-2 in the follicle, presumably in accordance with its function as a transport protein into the follicle. Consequently, there is an increased availability of free receptor-bound IGF-1 within the follicle. An additional explanation for the observed correlation may be that IGFBP-2 is synthesised independently of its systemic concentration in the follicle and is also proteolysed at the same time. It has already been demonstrated that IGFBP-2 is subject to significantly greater cleavage by proteases within the follicle than it is in plasma. Furthermore, this effect is enhanced by IGF-1 [77]. In addition to the reduced mRNA expression demonstrated in this study, increased proteolysis of IGFBP-2, as observed by Spicer et al. [78] in mares and Beg et al. [79], in cattle, may also influence the association between IGF-1 and IGFBP-2 in the follicle.

Moreover, a correlation was revealed between the expression of *IGF1R* mRNA and the concentration of IGFBP-F. No comparable correlation has been identified in other studies to date. The observed correlation indicates that with increased proteolysis of IGFBP in the follicle, resulting in elevated concentrations of IGFBP-F, there is a concomitant increase in the presence of IGF1R at the GC in the follicle. The proteolytic process affects the binding capacity of IGFBP, resulting in the cleavage of binding proteins and a subsequent alteration in the local concentration of receptor-available IGF-1 [80,81]. A simultaneous increase in *IGF1R* mRNA expression results in an enhancement of the effects of IGF-1 on follicular cells. Alternatively, it is possible that greater *IGF1R* mRNA expression creates more binding capacity for IGF-1, which in turn triggers greater synthesis of PAPP-A in the cells. In this regard, Myers et al. [82] demonstrated in humans that the addition of IGF-1 increased the proteolysis of IGFBP-4 in cell culture, irrespective of whether the IGF1R was previously blocked or not. This may indicate that the proteolysis of IGFBPs is independent of IGF-1 binding to the IGF1R, which is contrary to the correlation demonstrated here. Furthermore, this study did not find a correlation between IGF-1, *PAPP-A* mRNA expression, or *IGF1R* expression, which is consistent with the findings of Aad et al. [83]. In their study, they saw no effect on the abundance of *PAPP-A* mRNA after the addition of IGF-1 to in vitro cultured GC from small (1–5 mm) or large (8–22 mm) bovine follicles. In parallel, Santiago et al. [64] also found no association between IGF-1 in bovine FF and *PAPP-A* mRNA in GC. Therefore, it remains unclear what causes alliteration of IGFBP proteolysis by PAPP-A in the follicle. 

In terms of methodological limitations, the results of this study should be interpreted with caution. The measurement of mRNA expression in a relatively small number of samples may not fully capture the complex interactions within the system. To gain a deeper understanding of this process, it would be beneficial to analyse both the actual PAPP-A activity and the protein concentration of the IGF1R simultaneously. This approach would provide insights into the extent and dynamics of the potential interactions between protease and the IGF1R. 

## 5. Conclusions

In conclusion, the observed correlation between plasma and FF for IGF-1 and IGFBP-2 allows the hypothesis of a potential transfer between the endocrine somatotropic axis, the components of which are mainly produced systemically in the liver, and the local components of the IGF system in the follicle to be proposed. The association of blood BHB with systemic and follicular IGF-1 and blood IGFBPs highlights the influence of the metabolic state on these components. Moreover, the study indicates that the production and degradation of local IGFBP within the follicle, as evidenced by mRNA expression levels in granulosa cells, plays a pivotal role in modulating the IGF system. The results of this study do not provide a complete and definitive description of the origin of components of the IGF system acting locally on the follicle and oocyte. Furthermore, they are unable to identify the specific causal factors responsible for alterations in this system. In order to gain a comprehensive understanding of the local regulatory mechanisms, additional investigation beyond the scope of this study is necessary.

## Figures and Tables

**Figure 1 animals-14-02370-f001:**
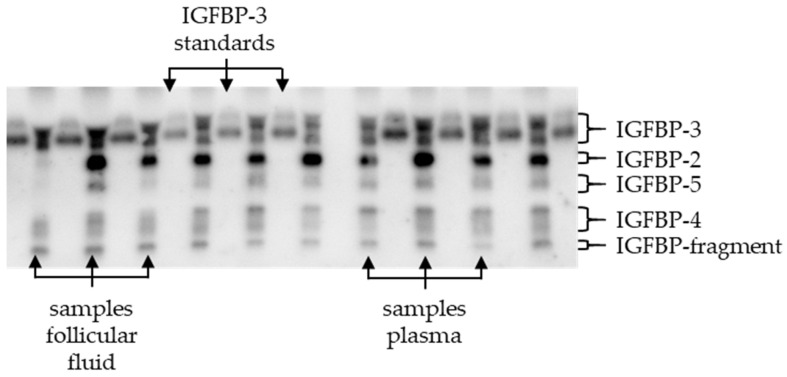
Example of a Western ligand blot with standards and samples; IGFBP = insulin-like growth factor binding protein, Gel no. 4_a_: representation of IGFBP-3 in the standard and the IGFBPs in the samples separated according to their molecular size; IGFBP in plasma and follicular fluid (IGFBP-2,-3,-4, and -5 and an IGFBP fragment that has bound biotinylated IGF-2), _a_: A21 results of the IGFBP analysis, Gel 141, 01.06.2022, Ligandis UG (Gülzow-Prüzen, Germany).

**Figure 2 animals-14-02370-f002:**
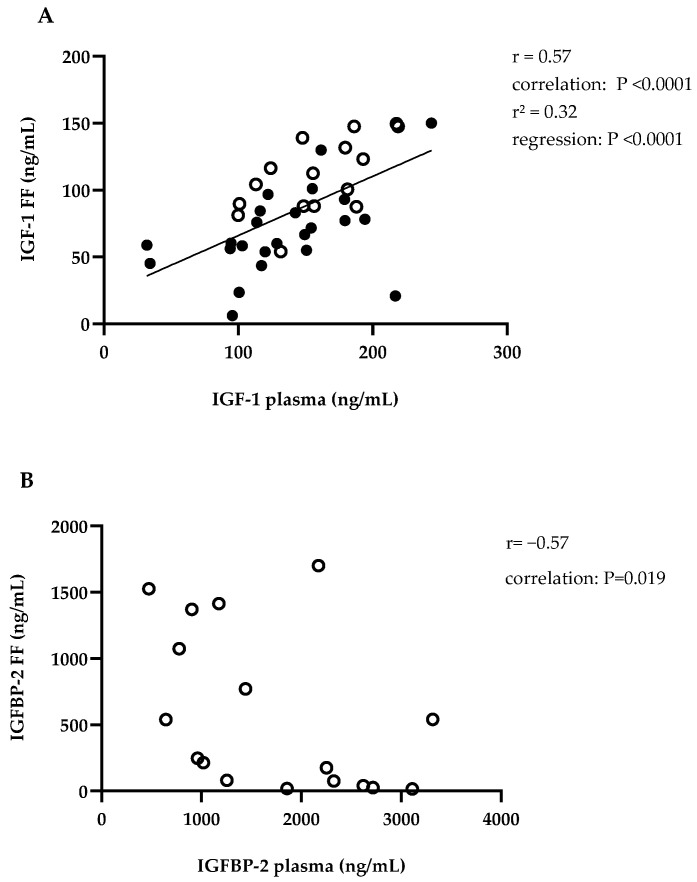
(**A**) IGF-1 in follicular fluid plotted against IGF-1 in plasma, IGF-1 = insulin-like growth factor 1, FF = follicular fluid, FF from study part I, ○, mean IGF-1 if more than one follicle/animal, *n* = 16, 109.64 ± 32.58 ng/mL, day 65 ± 16 p.p., FF from study part II, ●, *n* = 24, 68.79 ± 31.46 ng/mL, day 32 ± 3 p.p., Plasma from study part I, ○, *n* = 16, 158.90 ± 36.99 ng/mL, day 65 ± 16 p.p., plasma from study part 2 ●, *n* = 25, 133.27 ± 48.85 ng/mL, day 32 ± 3 p.p., with regression line; (**B**) IGFBP-2 in follicular fluid plotted against IGFBP-2 in plasma; IGFBP-2 = insulin-like growth factor binding protein 2; FF = follicular fluid, FF from study part I, mean IGFBP-2 if more than one follicle/animal, ○, *n* = 17, 578.4/1701/15.69 ng/mL, day 65 ± 16 p.p.; plasma from study part I, ○, *n* = 17, 1707 ± 904.7 ng/mL, day 65 ± 16 p.p.

**Figure 3 animals-14-02370-f003:**
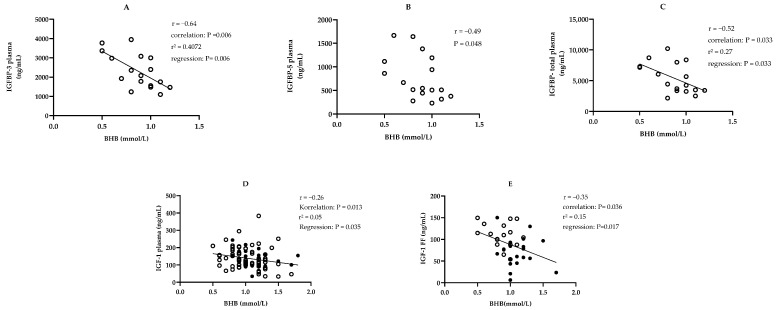
(**A**) IGFBP-3 and BHB in blood, study part I, IGFBP = insulin-like growth factor binding protein, BHB = β-hydroxybutyric acid, FF = follicular fluid, study part I, *n* = 17; with regression line, day 65 ± 16 postpartum (p.p.); (**B**) IGFBP-5 and BHB in blood; study part I, *n* = 17, day 65 ± 16 p.p.; (**C**) IGFBP-total and BHB in blood; study part I, *n* = 17; with regression line; day 65 ± 16 p.p; (**D**) BHB and IGF-1 in blood, study part I ○, *n* = 72, day 30 ± 2 p.p., study part II ●, *n* = 22, day 32 ± 3 p.p.; with regression line, nine outliers eliminated (BHB ≥ 2.2 mmol/L); (**E**) IGF-1 in FF and BHB in blood, study part I: ○, in FF mean IGF-1 if more than one follicle/animal, *n* = 17; study part II: ●, *n* = 20); with regression line, four outliers eliminated (BHB ≥ 1.8 mmol/L).

**Figure 4 animals-14-02370-f004:**
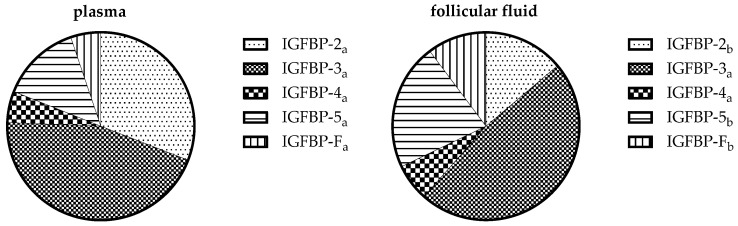
Shares of IGFBP in plasma and follicular fluid, study part I; IGFBP = insulin-like growth factor binding protein, FF = follicular fluid, study part I, in plasma (*n* = 17, IGFBP-2: 30.9/53/21%; IGFBP-3: 44.7 ± 8.2%; IGFBP-4: 5.4 ± 3%; IGFBP-5: 13.8 ± 3.1%; IGFBP-F: 5.2 ± 2.4%) and FF (*n* = 27, IGFBP-2: 13.8/36.3/0.5%; IGFBP-3: 47.6 ± 9.4%; IGFBP-4: 6.4/25.79/0.9%; IGFBP-5: 21.8 ± 8.6%; IGFBP-F: 10.27 ± 3.2%), percentage shares without a common letter (a, b) differ significantly between plasma and FF: IGFBP-5 (*p* = 0.001), IGFBP-2 (*p* < 0.001) and IGFBP-F (*p* < 0.001).

**Figure 5 animals-14-02370-f005:**
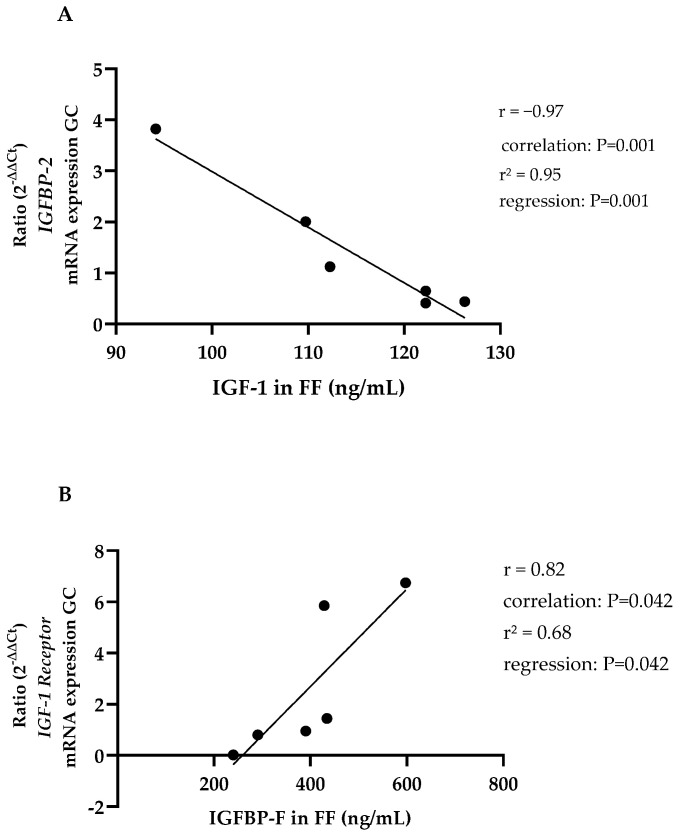
(**A**) Relative *IGFBP-2* gene expression in granulosa cells (GC) plotted against IGF-1 in follicular fluid, FF, and granulosa cells from study part I; *n* = 6; with regression line IGFBP = insulin-like growth factor binding protein; FF = follicular fluid. (**B**) Relative *IGF-1 receptor* gene expression in granulosa cells plotted against IGFBP-F in follicular fluid, FF, and granulosa cells from study part I; *n* = 6, with regression line; IGFBP = insulin-like growth factor binding protein; FF = follicular fluid.

**Table 1 animals-14-02370-t001:** Analysed parameters in number and type of samples per study part (study parts I and II) and time point (days postpartum).

Sample Type	Parameter	Samples			
		Study Part			
		I		II	
		*n*	Time Point	*n*	Time Point
blood					
	BHB	78	30 ± 2	*x*	
		77	50 ± 3	25	53 ± 3
		17	65 ± 16	25	32 ± 3
	IGF-1	78	30 ± 2	*x*	
		77	50 ± 3	*x*	
		17	65 ± 16	25	32 ± 3
	IGFBP	17	65 ± 16	*x*	
follicular fluid *^a^*					
	IGF-1	27	65 ± 16	25	32 ± 3
	IGFBP	27	65 ± 16	*x*	
granulosa cells					
	Gene expression	6	65 ± 16	*x*	

*x*: not assessed; *^a^* from all follicles in one animal; study part I: calculated mean; study part II: assessed in pooled follicular fluid.

**Table 3 animals-14-02370-t003:** IGFBP concentrations in plasma and follicular fluid on day 65 ± 16 postpartum, IGFBP = insulin-like growth factor binding protein, FF = follicular fluid; mean ± SD; median/95%/5% percentile, concentrations without a common letter (a–g) differ significantly, study part I, analysed using Western ligand blotting.

	IGFBP-Total (ng/mL)	IGFBP-2 (ng/mL)	IGFBP-3 (ng/mL)	IGFBP-4 (ng/mL)	IGFBP-5 (ng/mL)	IGFBP Fragment (ng/mL)
Plasma (*n* = 17)	5411 ^a^ ± 2467	1707 ^a,c,g^ ± 904.7	2312 ^a^ ± 895.7	175.5 ^b,d,f^/1061/54.0	545.3 ^b,c^ /1667/233	280.9 ^d^ ± 175.7
FF (*n* = 27)	4385 ^a^ ± 2356	362.7 ^b,e^/2512/16.4	1956 ^a,g^ ± 797.8	191.8 ^d,f^ /2937/23.5	863.9 ^b,g^ ± 362.4	406.8 ^b,d^ ± 149.4

**Table 4 animals-14-02370-t004:** Relative gene expression in different follicles; IGFBP = insulin-like growth factor binding protein, PAPP-A = pregnancy-associated plasma protein-A, IGF1R = IGF-1 receptor; follicles: size in mm, from study part I, day 65 ± 16 postpartum; ^1^, ^2^: follicles with the same number were punctured in the same animal; ^1^, ^2^: means without a common letter (a–b differ significantly).

Follicle Size (in mm)	IGFBP-2		IGFBP-4		PAPP-A		IGF-1 Receptor		18SrRNA and GAPDH
	Ct_IGFBP-2_	ΔCt *	Ct_IGFBP-4_	ΔCt *	Ct_PAPP-A_	ΔCt *	Ct_IGF-1 Receptor_	ΔCt *	Ct_RG_
12 ^1^	30.63 ^1^	15.06 ^1^	33.29 ^1^	17.72 ^1^	31.04 ^1^	15.47 ^1^	5.08 ^1^	10.81 ^1^	15.58 ^1^
9 ^1^	33.23 ^1^	14.96^1^	30.74 ^1^	12.47 ^1^	28.45 ^1^	10.18 ^1^	5.66 ^1^	7.94 ^1^	18.28 ^1^
18	31.69	13.60	34.01	15.92	30.14	12.05	8.60	9.96	18.10
20 ^2^	32.56 ^2^	11.83 ^2^	34.27 ^2^	13.54 ^2^	38.97 ^2^	18.24 ^2^	9.56 ^2^	15.93 ^2^	20.73 ^2^
9 ^2^	29.07 ^2^	12.76 ^2^	28.04 ^2^	11.73 ^2^	27.84 ^2^	11.53 ^2^	5.54 ^2^	7.73 ^2^	16.31 ^2^
12	30.47	14.39	28.22	12.14	26.63	10.55	5.89	10.55	16.08
mean	31.28 ^a^	13.76 ^b^	31.43 ^a^	13.92 ^b^	30.51 ^a^	13.00 ^b^	6.72 ^a^	10.48 ^b^	17.51
SD	0.62	0.52	1.16	0.98	1.81	1.30	0.76	1.21	0.79

* ΔCt = Ct_Gene_ − Ct_Reference gene_.

## Data Availability

The original contributions presented in the study are included in the article. Further enquiries can be directed to the corresponding author. The raw data supporting the conclusions of this article will be made available by the authors on request.

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
