# Peer review of "Association between IGF-1 and IGFBPs in Blood and Follicular Fluid in Dairy Cows Under Field Conditions"

_animals, 2024, doi:10.3390/ani14162370_

Round 1

Reviewer 1 Report

Comments and Suggestions for Authors

Insulin like growth factor 1 (IGF-1) plays a role in the endocrine and paracrine IGF systems of the ovaries in cows, affecting the development of follicles and oocytes. This study found that there is a potential transfer between endocrine IGF-1 and local components of the IGF system in follicles. This transfer may play a certain role in the connection between the metabolic status and fertility of cows.

However, Minor issue need to be addressed.

LINE 270: According to the description in Table 1, the sample size of follicular fluid in Table 4 should be 17, which is consistent with the sample size of plasma

Author Response

Thank you for your comments on the manuscript.

Please see the attachment for the point-by-point response.

Reviewer 2 Report

Comments and Suggestions for Authors

This manuscript summarizes results of a two-part in vivo study using dairy cows to evaluate the relationship between IGF1 and IGFBPs in blood and follicular fluid (FF). The study measured plasma/serum BHB, IGF1 and various IGFBPs (via western ligand blot) as well as mRNA (via RT-PCR) for the IGFBP2, IGFBP4, PAPPA and IGF1R in granulosa cells (GC) and ran correlations to see the relationship among them. IGFBP2 protein in plasma and FF were negatively correlated whereas IGF1 in plasma and FF was significantly correlated positively.  Serum BHB concentrations were significantly negatively correlated with plasma IGF1, IGFBP3, IGFBP5, IGFBP-tot, and FF IGF1.  Expression of GC IGFBP2 mRNA was negatively correlated with FF IGF1 and GC IGF1R mRNA was positively correlated with IGFBP-fragment. These results add to the plethora of published papers on the topic of IGF1 and IGFBPs in ovarian function in cattle. The authors did a nice job of discussing and summarizing their results to the related published papers. Table 2, 3A and 4 seems to be missing.

Main concerns/comments:

1.     Abstract:  It is unnecessary to add “r2” data to the abstract, as “r” says the same thing.

2.     Methods, line 132, 139:  The size range of follicles aspirated should be included here.

3.     Methods, line 145-146: has this hand-held device been validated for bovine serum? Is there a reference for this method.  Were duplicate readings conducted? At least the average coefficient of variation of replicate readings should be stated.

4.     Methods, line 149-154: Reference 29 is not easily accessible. So more details need to be added here. Were plasma samples extracted to remove IGFBPs? What is the assay sensitivity and intra-assay coefficient of variation?

5.     Methods, line 156-164:  the two references cited (30 & 31) cite other references for this procedure; please cite original papers that describe the method.  Also, for clarity, it should be stated that IGF2 was used as the ligand for IGFBP detection.  This can greatly affect interpretation of results as IGF1 and IGF2 more strongly bind to different IGFBPs.

6.     Statistics:  If data were not normally distributed, was there an attempt to homogenize the variance by using Log transformation or some other adjustment? Correlation analysis: Because some IGFBPs/hormones levels vary dramatically, the authors should consider also evaluating correlations of LOG (or natural Log) levels among variables before they firmly conclude no correlation exists.  The use of linear correlations assume a straight line but many biological variables change with a logarithmic-like increases and decreases (as seen in Fig. 2B).  

7.     Line 241 & 263: the range of follicle diameters should be stated here.

8.     Figure 5 A & B:  Y-axis labels need to include “granulosa cells” and “mRNA”. What is the unit of expression on these Y-axes?

9.     Line 426-428:  Reference 36 is not the correct reference, as it shows that insulin increases IGFBP2 mRNA in large follicle GC.  The correct reference is: Voge et al., 2004, Peptides 25:2195-2203 which shows that IGF1 increases IGFBP2 mRNA in large follicle GC but has no effect in small follicle GC.

 Additional comments:

1.     Line 43: delete “found”.

2.     Line 53-54:  add “…GH leads to the synthesis of some of….”   As GH does not stimulate synthesis of all of these; in fact GH decreases IGFBP2 synthesis. See: Radcliff et al., J Dairy Sci 2004 May;87(5):1229-35.

3.     Line 177: where is table A2?

4.     Line 181: here and elsewhere, use “min” for minutes.

5.     Table 2 appears to be labeled Table 3 in error.

6.     Line 187-188: Are there references that can be cited to support use of these reference genes?

7.     Line 291: give maximum diameter in >15 group.

8.     Line 292: give minimum diameter in < 15 group.

Author Response

(The authors gave the same response as above.)

Reviewer 3 Report

Comments and Suggestions for Authors

The current study investigates a potential link between the endocrine and paracrine systems through the examination of IGF-1 and related factors. While the study design is adequate, a major issue lies in its presentation. The authors should focus on presenting the data in a way that captures the readers’ attention. The article contains sufficient data to be considered for publication in the Animals journal; however, significant revisions are needed before acceptance.

The title is appropriate. However, the simple summary is overly lengthy and primarily focuses on scientific findings rather than presenting general facts for common readers. The introduction provides sufficient information but requires editing for better clarity and expression in English.

The Materials and Methods (M&M) section is poorly organized and lacks detail at several points. An ethical statement is missing, and the initial lines do not flow logically into subsequent sections. The study was conducted at multiple locations, necessitating clarification of their respective management practices. The term "Estrus Synch" should be used instead of "cycle synch."

On line 136, it is unclear which parameters were observed during the dry period.

The remaining laboratory analyses are acceptable but require English editing for clarity.

The Results section should clearly differentiate between Study I and Study II. The current presentation of results is difficult to follow.

In the figure legends, data do not need to be repeated. Similarly, Figure 3 is redundant as the percentages are already presented in a table. Ensure that tables include letters to indicate significance.

Lines 290-293 should be moved to the Materials and Methods section.

The Discussion section provides sufficient text and relevant citations; however, there are issues with English presentation.

The Conclusion should be revised to reflect the major findings of the study.

Comments on the Quality of English Language

The authors should seek the help of English Editor for revision. 

Author Response

(The authors gave the same response as above.)

Round 2

Reviewer 2 Report

Comments and Suggestions for Authors

This revised manuscript is much improved over the original version. A few additional points are listed below to further improve the manuscript:

1.     Line 65: delete “conducted by Rawan et al. (2015)” as ref. #16 is cited at end of sentence.

2.     Line 91: this should be “IGFBP” not “IGFPB”

3.     Regarding no follicle sizes recorded for study part II:  A statement should be added on line 163 stating “In study part II, all follicles > 3 mm in diameter were aspirated but individual diameters were not measured in study part II (29).”  As indicated in ref. #29.

4.     Line 217:  I think that this “Table 3A” should be Table A2” as I cannot locate where Table A2 is referenced in the text.

5.      Line 244-246: Based on the original delta delta publication (SEE: Livak KJ, Schmittgen TD. Analysis of relative gene expression data using real-time quantitative PCR and the 2(-Delta Delta C(T)) Method. Methods. 2001 Dec;25(4):402-8. doi: 10.1006/meth.2001.1262.), it is more commonly referred to as relative fold expression. Perhaps this original publication should also be cited.

6.     Lines 186-187, 414, 442-452: It seems that the reference numbers should be used in parentheses rather than the years.

7.     Table A2 should also state “Study part I” in the title.

8.     Results:  I can’t find anything said about PAPP-A mRNA results; the authors should make a statement summarizing PAPP-A results in the results.

9.     Discussion, line 517-534:  It is unclear why previous studies with bovine granulosa cells (GC) showing that IGF1 treatment in vitro does not affect PAPP-A mRNA in three experiments (see: Aad et al., 2006, Domest Anim Endocrinol. 31(4):357-72) and that no correlation existed between IGF1 in follicular fluid and GC PAPP-A mRNA in bovine follicles (see: Santiago et al., 2005, Domest Anim Endocrinol. 28(1):46-63). It seems that this section of the discussion would be more complete with these studies discussed.

Author Response

(The authors gave the same response as above.)
